# Therapy in the Course of Kidney Graft Rejection—Implications for the Cardiovascular System—A Systematic Review

**DOI:** 10.3390/life13071458

**Published:** 2023-06-27

**Authors:** Jakub Mizera, Justyna Pilch, Ugo Giordano, Magdalena Krajewska, Mirosław Banasik

**Affiliations:** 1Department of Nephrology and Transplantation Medicine, Wroclaw Medical University, 50-551 Wroclaw, Poland; justyna.pilch@student.umw.edu.pl (J.P.); magdalena.krajewska@umw.edu.pl (M.K.); miroslaw.banasik@umw.edu.pl (M.B.); 2University Clinical Hospital, Wroclaw Medical University, 50-551 Wroclaw, Poland; ugogiordano1@gmail.com

**Keywords:** cardiovascular side-effects, kidney rejection, Banff classification, nephrology, transplantology, cardiology

## Abstract

Kidney graft failure is not a homogenous disease and the Banff classification distinguishes several types of graft rejection. The maintenance of a transplant and the treatment of its failure require specific medications and differ due to the underlying molecular mechanism. As a consequence, patients suffering from different rejection types will experience distinct side-effects upon therapy. The review is focused on comparing treatment regimens as well as presenting the latest insights into innovative therapeutic approaches in patients with an ongoing active ABMR, chronic active ABMR, chronic ABMR, acute TCMR, chronic active TCMR, borderline and mixed rejection. Furthermore, the profile of cardiovascular adverse effects in relation to the applied therapy was subjected to scrutiny. Lastly, a detailed assessment and comparison of different approaches were conducted in order to identify those that are the most and least detrimental for patients suffering from kidney graft failure.

## 1. Introduction

Kidney transplantation is considered the gold standard treatment for end-stage renal disease, improving the quality of life and providing higher survival rates for patients. However, despite advancements in immunosuppressive therapy, kidney graft failure remains a significant complication [1]. Managing kidney transplant rejection involves administering a great variety of immunosuppressive agents such as steroids, calcineurin inhibitors and mycophenolate mofetil [2]. While these drugs can prolong the survival of the transplanted kidney, they can also cause adverse effects on the cardiovascular and endocrine systems. Moreover, patients who experience graft rejection are possibly at increased risk of cardiovascular events, including myocardial infarction and stroke [3]. Therefore, proper management of kidney graft rejection is not only crucial for renal function but also for reducing the risk of cardiovascular events [4].

The Banff classification points out several types of graft rejection based on the molecular background of the process and includes, most of all, antibody-mediated changes, T-cell-mediated changes and borderline rejection. This causes kidney graft rejection to be a heterogenous disease entity. In addition, it marks the possibility of the simultaneous occurrence of these rejection types. The molecular foundation of each type of graft rejection significantly alters the treatment recommendations, which has a substantial impact on the cardiovascular effects [5].

## 2. Materials and Methods

A thorough search of the literature was conducted. Authors included journals and articles published until May 2023. Using keyword variants such as the Banff classification, ABMR, TCMR, borderline rejection, mixed rejection, tacrolimus, steroids, cyclosporine A, mycophenolate mofetil, IVIG, plasma exchange, splenectomy, basiliximab, rituximab, everolimus, complement inhibitors, imlifidase, interleukin-6-inhibitors, kidney transplantation, kidney rejection and long-term cardiovascular implications after kidney transplantation, we obtained 1,464,795 citations from Google Scholar and PubMed databases, and 189,461 duplicate records were removed. Then, 7945 citations were screened on their abstracts and titles, and 6742 citations were excluded from further analysis. Subsequently, 1203 articles were found especially useful in our research. After a comprehensive analysis, we selected 125 citations that concerned the long-term influence of kidney graft rejection therapeutic regimens on the cardiovascular system (Figure 1). Articles written in English or Polish language were included, while articles written in other languages were ruled out.

## 3. Antibody-Mediated Rejection

Antibody-mediated rejection (ABMR) is a type of allograft loss in which kidney injury is caused by different pathogenic antibodies. These antibodies can be directed against human leukocyte antigens (HLAs), non-HLA antigens (i.e., angiotensin type 1 receptor) and blood-group antigens [6,7,8].

The consequences for the graft associated with injury mediated by donor-specific anti-HLA antibodies (DSA) were noted and described as early as 1953—in the first kidney transplant biopsy ever performed [9]. Almost 50 years later, during the Sixth Banff Conference on Allograft Pathology in 2001, the first international consensus diagnostic criteria and classification for ABMR in renal transplants were presented [10].

Since then, ABMR diagnostic criteria were revised several times in order to establish an optimal diagnostic scheme and improve the predictive value for allograft outcomes. The most recent ABMR diagnostic criteria and classification were issued in The Banff 2019 Kidney Meeting Report. Based on serologic evidence of DSA, histological features and clinical manifestations, antibody-mediated changes are subclassified into active ABMR, chronic active ABMR and chronic inactive ABMR [11,12]. The clarification of the diagnostic scheme along with the development of highly precise techniques for detecting anti-HLA antibodies has led to a significant increase in cases diagnosed with ABMR, making it a major cause of graft failure [13,14].

### 3.1. Active ABMR

The common term “active ABMR” comprises at least three diverse clinicopathologic forms: true acute ABMR, smoldering active ABMR and chronic active ABMR (CA ABMR) [12].

True acute ABMR occurs early post transplantation without previous chronic damage to the allograft. Typically, it manifests as acute graft failure in highly sensitized recipients having a memory humoral response. It can be reversed by following the current therapeutic scheme, primarily aimed at removing donor-specific antibodies with plasmapheresis, rituximab and intravenous immunoglobulin [15].

Smoldering active ABMR is usually diagnosed in patients with low-level DSAs, during surveillance or indication biopsy, while chronic active ABMR is considered to represent a continuum of the smoldering active form if it is not promptly diagnosed and treated. Hence, CA ABMR usually occurs in recipients with limited compliance. To prevent the increase in the number of patients developing the progression of smoldering active ABMR into CA ABMR, future clinical trials should focus on discovering novel agents intended for the treatment of active ABMR, other than (or in addition to) donor-specific antibodies removal [12] (Table 1).

### 3.2. Chronic Active ABMR

The category of CA ABMR includes lesions with mild activity and severe chronicity (e.g., g1, ptc1, cg3, ci3, ct3, C4d0), with severe activity and mild chronicity (e.g., g3, ptc3, cg1, ci0, ct0, C4d3) and intermediate ones. By consequence, a simple diagnosis of CA ABMR without specifying the severity of lesions based on different parameters does not give clinicians appropriate guidelines regarding treatment approach. This issue was recently highlighted in research conducted by the Banff Working Group, in which they found out that the ABMR diagnostic criteria and classification were interpreted in a highly variable way, which subsequently influenced therapeutic decision making and treatment outcomes [11] (Table 2).

### 3.3. Chronic (Inactive) ABMR

In order to diagnose chronic (inactive) ABMR, the following criteria must be met: (Table 3).

### 3.4. Evidence-Based Treatment Approaches in Active and Chronic Active ABMR

Despite ABMR being the main reason for kidney allograft rejection, there are still no approved therapies and effective, evidence-based treatment guidelines, due to an insufficient number of prospective randomized trials. To address this issue, The Transplantation Society gathered together international specialists who widely discussed the underlying pathomechanism of ABMR and the accuracy of current diagnostic criteria. While experts agreed that the main aims of therapy are to preserve graft function and reduce both histological injury and the titer of DSA, the evidence was not sufficient to unanimously support any particular therapeutic strategy. It is mostly caused by the fact that the vast majority of reports discussing the treatment of ABMR investigate small, heterogeneous patient populations and frequently include other types of allograft rejection (i.e., mixed antibody and TCMRs). Moreover, most studies followed the guidelines recommending the implementation of variable combinations of interventions, which consequently influences the final outcomes and makes it challenging to preclude meaningful conclusions about the effectiveness of different approaches [16,17]. Therefore, proposed recommendations for ABMR treatment are mostly based on expert opinion and clinical experience rather than evidence-based trials [13].

#### 3.4.1. Plasma Exchange and IVIG

Therapeutic approaches for ABMR have the overarching objectives of eliminating circulating DSA and decreasing DSA production. Although plasma exchange (PLEX) and IVIG are the strongholds of contemporary ABMR treatment, they do not have FDA endorsement. These treatment methods are commonly implemented to manage active ABMR; however, their modality, frequency and dosing regimens are subject to variability [12,16,17,18,19]. Despite a lack of robust supporting evidence, expert consensus at both the FDA Antibody-Mediated Rejection Workshop [20] and Kidney Disease: Improving Global Outcomes (KDIGO) [21] has deemed PLEX and IVIG as standards of care for acute active ABMR in view of their capacity to improve short-term results as proved in several research studies [15,22,23]. Nonetheless, evidence of their efficacy for long-term effects remains variable, necessitating the pursuit of new alternatives or adjunctive therapies for ABMR treatment. Furthermore, the number of PLEX procedures and IVIG dosing require better definitions.

#### 3.4.2. Complement Inhibitors

In recent years, researchers have shown growing interest in the complement system’s role in ABMR and have conducted several studies to explore the potential benefits of using various complement inhibitors in ABMR prevention and treatment. The primary objective of applying complement inhibitors is to avoid injury caused by DSA to the transplanted kidney [13].

Eculizumab is a monoclonal antibody targeting C5 and, by consequence, blocking the terminal complement pathway. According to a single-center study, eculizumab reduced the incidence of early active ABMR in HLA-incompatible transplants to 7% among treated patients from nearly 40% in historical controls. Another two multicenter randomized trials confirmed that eculizumab has a protective role against early active ABMR in positive crossmatch HLA-incompatible donors, both living [24] and deceased [25]. Despite these encouraging results, long-term follow-up studies have revealed that although eculizumab prevents the onset of early active ABMR, the long-term prevalence of chronic ABMR and graft survival is comparable to historical trials [26,27].

Proximal complement inhibition has also been explored as a potential therapeutic target. Berinert and Cinryze, two plasma C1 esterase inhibitors, have undergone testing in two pilot studies, indicating a possible improvement in allograft function in patients with ABMR [28,29].

#### 3.4.3. Rituximab

According to the KDIGO guidelines, Rituximab, a B-cell depleting agent, has been recommended as a treatment option for active ABMR [18]. Although Rituximab is commonly used for this purpose [30], its efficacy has not been definitively proven in three small randomized trials [16,19,31].

Another randomized, double-blind, multicenter prospective research study included 38 patients with active ABMR, all treated with PLEX, IVIG and steroids. The only difference in treatment strategy was whether recipients additionally received Rituximab or placebo. Ultimately, the study showed no significant difference in outcome between Rituximab and placebo groups, except for side-effects [32]. Similar conclusions were drawn by researchers in a prospective, randomized, double-blinded, placebo-controlled Spanish clinical trial where patients were treated either with IVIG plus Rituximab or IVIG plus saline infusion. In neither case were significant differences in outcome after 12 months were observed [33].

Despite these unpromising results of randomized controlled trials, several retrospective analyses have indicated some positive treatment effects of Rituximab combined with steroids, plasmapheresis and high-dose IVIG [13,17]. However, the optimal Rituximab dosage and number of therapy cycles that should be performed remains unclear, as well as the actual need for Rituximab as a multimodal regimen component [34].

#### 3.4.4. Imlifidase

Imlifidase (IdeS), which is currently undergoing clinical trials for ABMR treatment, has the potential to rapidly reduce and eliminate anti-HLA DSA [35]. It is an IgG-degrading enzyme of Streptococcus pyogenes, which acts by cleaving human IgG at a specific amino acid sequence within the hinge region responsible for producing Fc and F(ab)2 fragments that effectively block antibody-dependent cellular cytotoxicity [36]. While IdeS has been used safely for desensitization in highly sensitized individuals, its effectiveness in treating ABMR remains unclear, as patients often experience a rebound in anti-IdeS antibodies and DSA within a week of administration. The rebound may develop after 1 or 2 doses, consequently preventing repeated administrations. Therefore, IdeS is more likely to be used as an adjunct to other treatment approaches aiming to eliminate DSA in the long term rather than an isolated therapy for ABMR. However, the unique advantage of IdeS is that it enables highly sensitized patients to undergo transplantation quickly after donor identification, regardless of the crossmatch result [37].

#### 3.4.5. Interleukin-6 Inhibitors

A total of 36 patients suffering from chronic active ABMR who had not responded to the IVIG plus rituximab therapeutic scheme were included into a single-center, non-randomized trial of an anti-interleukin-6 receptor monoclonal antibody—tocilizumab. Tocilizumab administration resulted in a significant decrease in DSA levels and a stabilization in renal function. Moreover, both patient (91%) and graft (80%) survival rates at 6 years were superior when compared to historical trials. Based on these promising outcomes, other randomized control trials have been initiated with the aim to evaluate Clazakizumab (another anti-interleukin-6 monoclonal antibody) for the treatment of CA ABMR [38,39].

#### 3.4.6. Splenectomy

Surgical splenectomy, splenic radiation and splenic embolization performed as a rescue procedure for severe early ABMR have been documented in several case series. To be effective, all of the above-mentioned procedures must be executed quickly after the onset of early ABMR. However, evaluating the efficacy and effectiveness of splenectomy is difficult due to the challenges posed by patients, who, besides having undergone the surgery, are also known to be sensitized, have undergone desensitization therapy or have performed DSA [40,41].

## 4. T-Cell-Mediated Rejection

### 4.1. Acute TCMR—Treatment

Acute T-cell-mediated rejection is a pathological condition characterized by the activation of the recipient’s T cells in response to donor antigens presented by antigen-presenting cells (APCs) in the transplanted organ. The recognition of the foreign antigens by the T cells can occur through direct, semi-direct or indirect pathways, which ultimately results in the infiltration of the immune cells into the allograft and subsequent damage [42]. Acute TCMR usually can be diagnosed in a few weeks or months after the transplantation. Symptoms suggesting the diagnosis include proteinuria and the elevation of serum creatinine level, but the condition can be recognized by performing a biopsy and assessing histopathological scores.

The Banff classification points out several grades of acute TCMR and differentiates them by taking into account the occurrence of different types of interstitial inflammation of non-sclerotic cortical parenchyma (i), tubulitis (t) and intimal arteritis (v) [11] (Table 4).

The histological severity of acute TCMR is thought to be associated with inappropriate immunosuppressive treatment or impaired response to the applied medications, which, by consequence, yields worse graft outcomes [43].

Acute TCMR treatment differs between particular grades. Many guidelines recommend treating Banff grade I with steroids alone and Banff grade II with steroids including the addition of antithymocyte globulin (ATG) [44]. However, there are insufficient high-quality studies that provide histological support for this strategy, making it difficult to determine when patients need supplementary rejection treatment and when they do not require it. After the initial treatment of TCMR, some research suggests that the kidney function response to the treatment should guide further rejection treatment [43].

Moreover, the maintenance immunosuppression has to be included. Bouatou et al. suggested that acute TCMR treatment should consist of methylprednisolone pulses (500 mg/d for 3 days) and then oral prednisone tapered up to 3 months from the time of acute TCMR diagnosis to achieve 10 mg/d in all patients, in accordance with the centers’ protocols. Mycophenolate mofetil and tacrolimus with the trough level between 6 and 10 ng/mL can be used in the maintenance immunosuppression treatment. Some patients may require ATG administration (1.5 mg per kilogram per day, for 5 days) [45].

According to the analysis conducted by Aziz et al. on the group of 163 patients, histological responses to therapy are typically in accordance with the available recommendations to treat TCMR grade II—and potentially grade IB—with steroids and ATG. Moreover, they pointed out that the majority of patients with grade IA or IB responded to steroids alone, yet it was noticed that histological response did not correlate with kidney function responses [43].

Out of 87 patients who responded well to therapy as measured by kidney function, 5% had a partial response histologically and 9% presented no response. If the histology had been unavailable, none of these patients with a complete response by eGFR would have received additional care, but the protocol biopsy enabled the 14% of patients with a partial response or no response to do so. Furthermore, 68% of the patients with no response by eGFR who had available protocol biopsy findings were able to avoid needless treatment because the histology revealed a complete response [43].

Moreover, the biopsy findings enabled 74% of the patients with a partial response by eGFR to forgo further treatment because the histology revealed a complete response. With a second cycle of steroids, 34 patients who had a partial response or no response histologically after the initial rejection treatment experienced a 79% response rate, suggesting that a partial response or no response histologically after initial treatment can typically be reversed [43].

It seems that acute cellular rejection in kidney allografts has well-established diagnostic criteria and treatment methods [46,47]. According to a review by Brennan and Malone, a reduction in serum creatinine should be taken into consideration as the threshold for a successful treatment of rejection [44]. The research also marks the need of careful monitoring of serum creatinine levels after the treatment of rejection [44].

### 4.2. Chronic Active TCMR—Treatment

Chronic active T-cell-mediated kidney rejection (CA TCMR) is a type of kidney allogenic graft rejection reported for the very first time at the XIII Banff Conference on Allograft Pathology in 2015 [48]. Nowadays, CA TCMR can be diagnosed and graded on the basis of histopathological criteria established at the Banff 2019 conference. The elementary lesion present in CA TCMR is inflammation in areas of the cortex with interstitial fibrosis and tubular atrophy (i-IFTA), but interstitial inflammation of the total cortex (ti) and tubulitis (t) are also important in establishing the diagnosis [11].

Loupy et al. suggest that CA TCMR should be staged on grades IA, IB and II [11]: (Table 5).

Moreover, in grades IA and IB taking i-IFTA into consideration, other possible causes of these lesions have to be ruled out to establish a proper diagnosis.

As CA TCMR is a newly described term, there is a lack of clear guidelines for the management of the ailment. Studies have shown that some researchers hesitate whether CA TCMR should be treated or even diagnosed [49]. In 2017, more than 90% of cooperating clinical centers diagnosed CA TCMR and treated the ailment with immunosuppressive medications including steroids in more than 80% of cases. Yet, the group was not homogenous; only in 36% of these cases did doctors treat CA TCMR routinely. A total of 49% treated the disease under the condition that the therapy depended on the Banff (i and v) lesion scores, the lack of severe IFTA and on the cause of biopsy. Furthermore, 15% of clinicians treated CA TCMR rarely. The main reasons pointed out by physicians behind their hesitation concerning whether to treat patients were the lack of clinical data and the belief that possible benefits are outweighed by the risks of the therapy. Importantly, in 9% of clinical centers, CA TCMR was not diagnosed at all [12].

Recent scientific findings suggest that the application of methylprednisolone (4 mg/d), mycophenolate mofetil (500–1500 mg/d), Tacrolimus (minimal concentration 5–8 th/mL), everolimus (minimal concentration 3–8 ng/mL), basiliximab (20 mg in days 0 and 4) and ATG may be beneficial for patients suffering from CA TCMR. The clinical case report performed on a modest group of three patients where the combination of these drugs was administered gave promiscuous results. Moreover, none of them demonstrated deterioration of eGFR [50].

On the other hand, different studies determined that only in a small subset of cases is it possible to improve kidney function in patients suffering from CA TCMR. In that research, methylprednisolone 500 mg was administered intravenously for three days, followed by oral prednisone 5 mg daily for more than four weeks. Moreover, if the stage of CA TCMR was IB, physicians provided patients not only with steroids but also included ATG (1.5 mg/kg) daily for 4 days. At 4 weeks following biopsy, only 20% of patients had at least 50% of their eGFR recovered. The stage of CA TCMR and treatment response were found to be related. Patients with grade IB of CA TCMR tended to respond less favorably than those with grade IA. The observed response rate was higher in patients who did not fit the criteria for co-existing acute TCMR, indicating that there is no connection between treatment response and the management of acute TCMR. Patients with severe, moderate or mild parenchymal scarring in the biopsies demonstrated no significant differences. However, in some circumstances where the criteria for acute TCMR are not met and substantial parenchymal scarring is visible, additional immunosuppressive medication may be contraindicated [51].

There is still a need for large, randomized studies where patients would be treated with different immunosuppressive agents and combinations of those medications to work out an ideal scheme for CA TCMR treatment. Even though some clinical research has already been conducted, the results are questionable and uncertain [49].

## 5. Borderline Rejection—Treatment

Borderline cellular rejection (BCR) is part of the Banff classification system. According to the 2019 Banff classification, in order to be diagnosed with borderline (suspicious) for acute T-cell-mediated rejection (TCMR), the criteria in Table 6 have to be met [11].

The Borderline TCMR (BL-R) encompasses a broad spectrum of cases, ranging from mild inflammation that barely meets the diagnostic threshold of 10% with only one inflamed tubule, to acute renal dysfunction with intense inflammation and tubulitis, characterized by 24% to 49% inflammation as i1/t1 or i2/t1. The interpretation of BL-R is uncertain and can vary from a state of “very mild rejection” that does not necessitate any medical intervention, to a condition that is “suspicious for rejection” and is assumed to be TCMR, necessitating immediate treatment [52]. Lesions described as mild tubulitis (Banff t1) with inflammation involving >25% of the nonsclerotic cortex (Banff i2/3) and no intimal arteritis are also included in the borderline category, the same as in the previous versions of the Banff classification. However, biopsies without inflammation in at least 10% of the nonsclerotic cortex (i0), regardless of being classified as tubulitis t > 0, are no longer included in this category. The same consideration is valid for cases with inflammation in the absence of tubulitis (i > 0, t0). All of those findings have simply to be mentioned in the description of the biopsy [11].

Back in 2016, a worldwide web-based survey among 503 members of the Renal Pathology Society was conducted, which concerned an ongoing debate about the minimum threshold for tubulointerstitial infiltrates (Banff i) in the Borderline category. The survey was sent out to 503 members of the organization, of whom 153 responded. Out of the 138 respondents using Banff Borderline:

39 (28%) considered i0 to be the minimum threshold;

92 (67%) required i1 (10–25% of parenchyma inflamed);

7 (5%) used variable thresholds depending on the clinical circumstances [53].

The results of this survey-based research are in line with the diagnostic criteria included in the 2019 Banff classification.

Even in patients who underwent kidney transplantation (KT) with low immunological risk and sustained graft function, subclinical inflammation, including borderline lesions (BL), is frequently discovered. Interstitial fibrosis and tubular atrophy (IFTA), worsening of the graft function and loss of the KT can all result from these lesions, particularly when interstitial inflammation and tubulitis occur. The efficacy of treating BL lesions, however, is not well established [54].

In recent research carried out between 2018 and 2020, the outcomes of the treatment of acute BCR have been evaluated thoroughly. All patients diagnosed with BCR were treated with a rapid steroid regimen, followed by a biopsy to observe any histological changes due to therapy. The treatment that has been used consisted of the following:

A rapid steroid regimen—methylprednisolone 500 mg IV once a day for 3 days followed by 250 mg IV once a day on day 4, 125 mg IV once a day on day 5, start oral prednisone at 60 mg on day 6 and taper over 1 week by reducing the dose by 10 mg daily to the maintenance dose of 5–10 mg daily.

An induction protocol is based on using basiliximab, alemtuzumab or thymoglobulin.

Basiliximab is an anti-interleukin-2 receptor (anti-IL-2R) monoclonal antibody that functions as an antagonist of the CD-25 alpha receptor on activated T-cells. It has been integrated into numerous immunosuppressive therapies alongside cyclosporin A, tacrolimus, belatacept or azathioprine as part of the induction phase, immediately after transplantation [55,56,57]. Owing to the high affinity that Basiliximab has for the alpha subunit of the IL-2 receptor (IL-2Rα), it impedes IL-2 from binding, thus inhibiting activated T-cells from further proliferation [58]. In addition, its monoclonal nature results in a more predictable and consequently safer immunosuppressive effect [59]. Nonetheless, in a patient with an ongoing acute rejection, the IL-2Rα is no longer expressed on the surface of T-cells, which limits the use of these antibodies [60].

Alemtuzumab is a humanized IgG1 monoclonal antibody directed against CD52, which causes a profound depletion of immunocompetent cells lasting from 1 to 2 months for natural killer (NK) cells and monocytes, to 6 to 12 months for T and B cells [61,62]. The above-mentioned antigen is a glycoprotein expressed on the surface of mononuclear cells including NK cells, and T and B lymphocytes, but also on the lining of the male reproductive tract; however, its function remains unclear [63]. Alemtuzumab has been employed in hematologic malignancies, as an immunosuppressive agent for autoimmune disorders, and in bone marrow and solid organ transplants. Nonetheless, regulatory clearance for its use has been granted only in B-cell chronic lymphocytic leukemia (B-CLL) [64].

Patients were included in the responsive group when the follow-up biopsy had interstitial inflammation involving less than 10% of the nonsclerotic cortex (i = 0), with no tubulitis (t = 0). The persistence of acute BCR histological changes on the follow-up biopsies conducted after completing the rapid steroid regimen was compared. It was found that persistent acute BCR occurred in 63 out of 91 patients despite treatment; thus, only a 31% response rate was achieved. There were no differences between the two groups in terms of induction therapy with alemtuzumab, thymoglobulin and basiliximab [65].

In a study conducted between 2008 and 2015, Dale et al. analyzed the response to steroid-based therapy among 90 patients who had been diagnosed with acute BCR based on biopsy. It was a single-center, observational retrospective cohort study of adult (≥18 years old) kidney transplant recipients found to have i0 borderline infiltrates suspicious for acute TCMR. According to the 2005 Banff classification, i0 was defined as less than 10% interstitial inflammation. The regimen constituted 500 mg of intravenous (IV) methylprednisolone per day for the first three days, followed by 250 mg on day 4, 125 mg on day 5 and 75 mg on day 6. Then, the patients were supposed to be administered oral prednisone tapering from 20 mg per day to 5 mg per day over the subsequent 2 weeks. Subsequently, in the follow-up biopsies, it was found that less than half of the patients resulted as negative for rejection. The findings strongly suggest that in relatively low-immunologic-risk patients, the development of i0 borderline infiltrates is not affected by corticosteroid treatment [66].

A single-center, cross-sectional cohort study compared the functional and histological outcomes of consecutive BL-R diagnoses (*n* = 146) against normal controls (*n* = 826) and acute TCMR (*n* = 55) from 551 renal transplant recipients. Acute glomerulitis, inflammation, tubulitis, peritubular capillaritis, the presence of acute tubular necrosis (ATN), chronic tubular atrophy and fibrosis are findings that were associated with BL-R. A total of 72.6% of TCMR cases were diagnosed as borderline rejection, which revealed itself as the most frequently occurring phenotype, exceeding the incidence of classical acute TCMR (A-TCMR). BL-R was treated in 65.8% of episodes (96/146, vs. 96.4% for A-TCMR, 53/55, *p* < 0.001), using pulse methylprednisolone (*n* = 94), anti-lymphocyte globulin (*n* = 12), intravenous immunoglobulin (*n* = 7) and/or upscaled maintenance immunosuppression (*n* = 42). A dismaying outcome that emerged during the study was the inadequate efficacy of traditional anti-T-cell therapy, which entailed administering pulse corticosteroids, in expeditiously managing BL-R (as well as A-TCMR) [52].

Currently, in addition to the uncertainty regarding the clinical impact of treatment for BCR, treatment modality, regimen and approach are also subjects of discussion. A clinical practice survey carried out in Canada in 2018, which polled 47 transplant providers, revealed that all participants chose to address borderline rejection, albeit with varying methods and approaches [67,68]. The results of the above-mentioned research have uncovered the need for fulfilling long-term comparative studies to evaluate whether the diagnosis of acute BCR necessitates a pharmacological therapy and, if so, which one would be the most reasonable to opt for. New findings in this area may play a major role in protecting affected patients from further immunosuppression that may be harmful and have no beneficial impact on the outcomes.

## 6. Mixed Rejection—Treatment

It is crucial to acknowledge that TCMR and ABMR are not necessarily separate diagnoses. In fact, Feucht’s study, which greatly influenced the current literature concerning ABMR, demonstrated the presence of C4d staining in biopsies with TCMR [69]. Depending on the transplant center, the occurrence of mixed TCMR-ABMR seems to vary, with the lowest approximations suggesting around 6% [70,71]. However, in a series of 67 ABMR biopsies carried out in Norway, early and late ABMRs biopsies had concurrent TCMR in 63% and 96% of cases, respectively, when borderline changes were taken into consideration in the analysis. If only TCMR Banff grade IA or higher would have been analyzed, 43% of early and 48% of late biopsies met the criteria for mixed rejection [72].

Determining the true incidence of mixed TCMR-ABMR is complicated by the presence of inflammation and tubulitis in areas with scarring. In such cases, a strict application of the Banff Schema may result in TCMR being underdiagnosed in at least 33% of biopsies [73].

Regarding the pathogenesis of mixed rejection, it is impossible to ascertain from a single biopsy whether TCMR exposed epitopes, leading to the development of antibodies, or ABMR occurred in the first place, followed by a subsequent influx of T cells [74]. Clinically, it has been observed that one or more episodes of TCMR can precede the development of ABMR [75,76], some of which may be subclinical [67,68,77]. However, there is currently a lack of concrete data to determine the frequency of antibodies preceding T-cell influx, and vice versa. It is probable that both scenarios occur, but their relative frequency remains unknown [78].

The wide range of incidence of mixed TCMR-ABMR reflects differences in diagnostic criteria and suggests that the further standardization of those is necessary. In patients who suffer initially from either T-cell- or antibody-mediated injury, the development of mixed TCMR-ABMR can substantially aggravate the clinical course. More research is required to determine the incidence at which these two pathologies co-occur [78].

Clinical monitoring and therapy centered around T cells remain relevant for the prevention of both cellular and humoral rejection, the management of steroid-resistant TCMR (which affects up to 20% of patients) and the optimization of clinical outcomes in mixed TCMR-ABMR. The latter is more common than generally acknowledged and continues to be linked with an unacceptably high rate of graft loss [78]. Chronic active ABMR is the most well-recognized cause of graft failure [17], while TCMR usually coexists with a mixed rejection phenotype [79]. Patients who have been found with donor-specific antibodies (DSAs) are susceptible to mixed ABMR-TCMR, which is prevalent in numerous studies and may necessitate a combined anti-T-cell and anti-B-cell approach for optimal results [78].

A prospective study examined the effectiveness of treating non-sensitized renal allograft recipients with tacrolimus (Tac) and mycophenolate mofetil (MMF) alone, without immunoadsorption (IA) or plasmapheresis (PPH), for early acute humoral rejection (AHR), defined as AHR occurring within two weeks. During the first two weeks following transplantation, 11 out of 160 (7%) Chinese renal allograft patients had AHR combined with cellular rejection (varying from Banff borderline to Banff IIB), and they all underwent Tac-MMF treatment. Its incidence rate is similar to that seen in groups that are not Asian [80].

During the study, the following immunosuppressive protocols were used:Cyclosporine A (CsA), MMF and steroids;Tac, MMF and steroids.

Applied regimen:MMF—the initial dose was 1.5 g per day.Calcineurin inhibitors—administered when the serum creatinine level decreased to 50% of pre-transplant levels. The initial dose of Tac was 0.6 mg/kg per day, while CsA was initiated at 4 mg/kg per day. Both were gradually raised in accordance with the restoration of graft function. Tac and CsA maintenance doses were adjusted to trough levels: 6–12 ng/mL for Tac during the first 6 months, followed by 4–8 ng/mL for the next 6 months; 150–250 ng/mL for CsA during the first 6 months, followed by 100–200 ng/mL for the following 6 months.Standard corticosteroid tapering—methylprednisolone IV (500 mg) on days 0–2, followed by oral prednisone 80 mg/day on day 3, which was then decreased to 10 mg/day increments to 20 mg/day. The dose of corticosteroid was then reduced slowly to 5 mg/day.

According to the above-mentioned research, Tac coupled with MMF are sufficient to stop early mixed cellular and humoral C4d-positive rejection in recipients of non-sensitized renal allografts [80].

In summary, the use of Tac and MMF together, without the inclusion of PPH or IA, appears to be a promising and economical approach for the treatment of early mixed cellular and humoral C4d-positive rejection in renal allograft recipients. This holds true, at least for non-sensitized patients without any signs of anti-endothelial cell antibodies. However, its effectiveness in sensitized cases and “pure” AHR is yet to be determined. It is worth noting that previous research has indicated that less than 5% of cases of C4d-positive AHR can be categorized as “pure humoral rejection” [81]. Thus, a comprehensive multicenter study would be beneficial in validating these findings.

## 7. Long-Term Implications for the Cardiovascular System Caused by Treatment Methods Applied in Kidney Rejection Treatment

As previously mentioned, the underlying pathomechanisms of kidney graft rejection can vary significantly, leading to considerable heterogeneity among therapeutic regimens. Depending on the chosen treatment protocol, the range of cardiovascular adverse effects manifestation can differ.

### 7.1. Prevention of Acute Graft Rejection

A paramount concern in renal transplantation is to prevent acute graft rejection. The current standard of care involves administering a calcineurin inhibitor (CNI), either tacrolimus or cyclosporine A (CsA), combined with mycophenolate mofetil (MMF) and steroids. However, the use of this regimen is associated with a considerable (approximately 20%) incidence of acute rejection at 1 year and the substantial toxicities of immunosuppressive agents [82].

Cyclosporine A (CsA) and tacrolimus are calcineurin inhibitors, considered to have remarkable efficacy in diminishing the incidence of acute graft rejection after transplantation. Nevertheless, continuous treatment with cyclosporine may result in an accelerated deterioration in graft function due to its nephrotoxicity. Moreover, the occurrence of other cyclosporine-induced adverse effects, including hypertension and hyperlipidemia, contributes to increased cardiovascular morbidity in renal transplant recipients [83]. Apart from elevated blood pressure, disadvantageous effects of CsA resulting in an increased cardiovascular risk profile are mainly related to metabolic cardiovascular risk factors: a quantitative increase in low-density lipoprotein (LDL) particles, an increased oxidizability of LDL particles, and increased homocysteine and plasma lipoprotein(a) (Lp[a]) levels. Additionally, cyclosporine has been reported to present unfavorable effects on the fibrinolytic system [84].

Although tacrolimus is also associated with nephrotoxicity, several studies have reported that its administration as the primary immunosuppressive agent enhances graft survival rates in patients [85,86]. Compared to cyclosporine, tacrolimus is described to have even more potent immunosuppressive activity and, concurrently, a lower propensity for inducing hypertension and hyperlipidemia, thereby improving the cardiovascular risk profile. In a randomized controlled study conducted by Artz et al., the conversion from cyclosporine to tacrolimus resulted in a sustained decrease in both systolic and diastolic blood pressure, and a consistent enhancement in the serum lipid profile (total and LDL cholesterol levels, triglyceride levels, the oxidizability of the LDL particles and fibrinogen levels), ultimately leading to a decrease in the Framingham risk score used to estimate the 10-year risk of developing cardiovascular disease. However, the significant drawback of tacrolimus is an elevated risk of developing diabetes mellitus, which may counteract the beneficial effects of decreased blood pressure and improved hyperlipidemia on cardiovascular morbidity and mortality [83].

Since the concomitant administration of a CNI and steroids aggravates the increase in cardiovascular risk after renal transplantations, several recent trials have investigated the efficacy and safety of the complete removal or cessation of CNIs or steroids subsequent to the transplantation [82].

Mycophenolate mofetil (MMF) is mainly used in association with other immunosuppressive agents with the aim to prevent graft rejection. As mentioned above, MMF administration was also proved to be beneficial in the treatment of CA TCMR [50]. Since it is classified as an immunosuppressive medication, the most serious and common side-effects are related to the higher risk of cancer development (especially lymphomas and skin cancers) and opportunistic infections. Adverse effects related to the cardiovascular system include tachycardia, hypotension or hypertension and vessels’ dilatation (frequency ≥1/100 and <1/10 patients) [87]. The combination of MMF with tacrolimus has become the most frequently implemented regimen following kidney transplantation. This therapeutic approach is distinguished by its efficacy in preventing both clinical and subclinical acute rejection, while concurrently affording a favorable tolerability profile. Tacrolimus/MMF displays a propitious cardiovascular risk profile in comparison to other immunosuppressive regimens, such as steroids/tacrolimus or steroids/tacrolimus/azathioprine. Steroid-sparing approaches yield a significant alteration in the cardiovascular risk profile, with noticeable benefits observed especially in parameters such as arterial hypertension, lipid profile and glucose metabolism disorders. However, it is yet to be established whether the noted differences in cardiovascular risk factors will ultimately lead to a diminished incidence of cardiovascular events in the long-term follow-up [88].

### 7.2. ABMR

Although ABMR treatment guidelines are mainly based on experts’ opinion and clinicians’ experience, plasma exchange and IVIG are considered the standard of care.

Plasmapheresis is a relatively low-risk procedure; yet, in a small subset of cases (0.025–0.2%), it may be associated with life-threatening complications. One of the most common disorders include all events related to the loss of calcium ions from the plasma and can be manifested by hypotension, reduced cardiac output, as well as ventricular arrhythmias. Hypocalcemia is thought to be caused by the interaction of the anticoagulants used in plasmapheresis with calcium ions. In addition, due to the removal of coagulation factors such as V, VII, IX, X, fibrinogen and antithrombin during the procedure, patients may exhibit hemorrhagic diathesis or states of hypercoagulability [89]. IVIG is also generally regarded as a safe medication. However, severe side-effects may occur during therapy, particularly in elderly patients with multiple cardiovascular risk factors or those with pre-existing renal failure [90] Clinical trials have identified a correlation between IVIG administration and thromboembolic events such as myocardial infarction, stroke, pulmonary embolism and deep vein thrombosis. The incidence of these events is believed to be influenced by the patient’s underlying medical condition, as well as the increase in blood viscosity that follows the administration of high doses of IVIG [91]. Therefore, close monitoring of patients is essential following renal transplantation, where this method of treatment is applied [92].

Rituximab is an anti-CD20 monoclonal antibody that is strongly recommended in ABMR treatment. It is estimated that side-effects upon the therapy using rituximab can emerge in approximately 87% of patients and mostly include fever, chills or rigors. Yet, few can experience cardiovascular complications including arrhythmias, trigeminy, irregular pulse, monomorphic VT or supraventricular tachycardia [93]. Moreover, rituximab is thought to increase ventricular dysfunction rates in a mechanism of elevation of TGF-b levels and reticuline fiber formation in cardiac muscle. As a consequence, it may affect cardiac contractility and conduction [94]. Rituximab’s therapeutic action involves cell lysis through complement-dependent cytotoxicity, antibody-dependent cellular cytotoxicity, and apoptosis. The CD20 antigen is found on immune-effector cells and, after cytotoxic lysis, it may deposit itself in other tissues such as cardiac myocytes. It is conceivable that the drug affects conduction by inhibiting the CD20 antigen’s calcium-ion-channel properties. Calcium-ion channel inhibition in cardiac myocytes may result in the development of early after-depolarizations. If these early after-depolarizations reach a certain amplitude, they may be manifested by ventricular extrasystole or polymorphic VT (including torsades de pointes). Rituximab has been shown to prolong the QT interval, potentially leading to a variety of arrhythmias [93,95]. Rituximab has also been found to interact with COVID-19 vaccines, potentially leading to pancytopenia, iatrogenic cytokine release syndrome or coagulopathy. [96]. Given the risk of rituximab to affect the safety and efficacy of vaccines recommended for kidney transplant patients, the timing of rituximab treatment in relation to COVID-19 vaccination uptake should be carefully considered [96,97].

Since plasma exchange together with IVIG and rituximab do not provide patients with 100% treatment effectiveness, alternative approaches are being considered. Recent scientific findings put the spotlight on particular eculizumab (anti-C5 monoclonal antibody) and C1-esterase inhibitors, which belong to the group of complement inhibitors [24]. In the study performed by Marks et al., the efficacy and safety of eculizumab were compared to the standard of care in patients suffering from ABMR. Although eculizumab application may be associated with vascular disorders (15.7%) such as deep vein thrombosis (3.9%), the incidence of these events is low and remains in most cases very similar to the standard of care. Yet, deep vein thrombosis was not observed in standard-of-care methods. However, the group of examined patients was too small to determine whether the findings, in this case, are pertinent [24]. C1-esterase inhibitors were also tested for potential beneficial influence on the cardiovascular system. The findings imply that they may be able to prevent thrombus formation and inhibit coagulation [98].

Another innovative approach includes the application of imlifidase, which belongs to the group of cysteine proteases. Its mechanism of action relies on the degradation of the recipient’s IgG. Even though the medication gives promiscuous outcomes, especially in desensitization treatment, it is not devoid of side-effects on the cardiovascular system [99]. Clinical trials imply that imlifidase can often (in more than 1/100 but not more than 1/10 of patients) cause hypo- or hypertension and sinus tachycardia events. The medication is subject to additional monitoring, and its use should be carefully evaluated in patients with cardiovascular risk due to the potential for adverse effects [100]. Furthermore, tocilizumab and clazakizumab, which both rely on the inhibition of interleukin-6, may be useful, especially for patients suffering from ABMR resistant to IVIG and rituximab treatment. Since COVID-19 is a treatment indication for tocilizumab, it has been extensively studied, especially in this matter. Hence, much of the information that is currently accessible focuses on side-effects in these patients. Nonetheless, it has been proven that tocilizumab administration causes an aberrant lipid profile. Due to the increased risk of hypertriglyceridemia and hypercholesterolemia, patients are advised to perform a fasting lipid profile 2–3 times during the first six months of treatment and then once a year. If dyslipidemia is confirmed, statins treatment may be advised [101,102]. Moreover, tocilizumab characteristics imply that common side-effects upon the therapy include hypertension and hypofibrinogenemia, which may lead to coagulopathies. The medication can cause increased weight, which is a risk factor for cardiovascular diseases.

Some studies suggest the role of splenectomy as a rescue procedure for severe early ABMR. Yet, overall splenic preservation has become more prevalent recently in an effort to reduce the risk of long-term cardiovascular disorders caused by the procedure itself [103]. The splenectomy performed in patients with hematological indications was associated with thromboembolic complications in up to 10% of them. The events can range from portal vein thrombosis to deep vein thrombosis or pulmonary embolism and may result from a higher-than-average platelet count following splenectomy, which reaches maximum levels by day 10 after the procedure. As long-term side-effects upon the splenectomy, cases of pulmonary hypertension, enhanced arteriosclerosis or arterial thrombosis have also been reported [103].

### 7.3. TCMR

In contrast to ABMR, more extensive investigations and research were conducted on the treatment recommendations for TCMR. Current guidelines recommend administering steroids solely in the therapy of Banff acute TCMR grade I, while Banff acute TCMR grade II should be treated with steroids combined with anti-thymocyte globulin (ATG) [44]. There are numerous therapeutic agents within the group of steroids, but prednisone and methylprednisolone are the primary medications used in the treatment of kidney rejection. Steroids are known to have well-defined and recognizable adverse effects during therapy; however, the incidence and severity of these effects are usually dependent on the dose and duration of treatment [104]. Protocols mentioned in our study indicate that acute TCMR should be treated with methylprednisolone pulses (500 mg/d for 3 days), then oral prednisone tapered up to 3 months and CA TCMR with methylprednisolone 500 mg administered intravenously for three days, followed by oral prednisone 5 mg daily for more than four weeks. Patients provided with steroids treatment most frequently report weight gain, bruising and oedema. Moreover, dyslipidemia and hypertension can be detected. Diabetes mellitus, which—along with weight gain and hypertension—may increase the risk of cardiovascular events, is another well-known toxicity of glucocorticoids. Patients should be well monitored for the occurrence of metabolic syndrome [105]. Another side-effect upon the therapy includes hypernatremia, which can lead to water retention and, by consequence, to heart failure, so a salt reduction diet should be prescribed. Moreover, ionic disorders may be the cause of arrhythmias. The cases of life-threatening arrhythmias including cardiac arrest were observed after quick (under 10 min) administration of significant doses of methylprednisolone (at least 500 mg) intravenously [106]. Taking into consideration the population of patients after renal transplantation, studies imply that 15% of them will develop steroid-related hypertension and, in approx. 10% of cases, posttransplant diabetes mellitus will occur [107]. Yet, some clinical trials imply that small, fixed daily doses (5 mg) of prednisone do not increase the incidence of diabetes or hypertension, but small changes in lipid profile and weight gain were noticed [108].

As mentioned above, ATG is advised for patients suffering from acute TCMR or chronic active TCMR (1.5 mg per kilogram per day, for 4–5 days). It is also useful in induction protocols in patients with borderline rejection. The characteristics of ATG suggest that the medication can often cause hypotonia in patients. Less common but potentially severe adverse reactions reported to occur after ATG uptake include acute thromboses [109,110]. In general, taking into consideration only cardiovascular side-effects and excluding immunological complications, ATG seems to be acceptably safe for patients [111,112].

Recommendations for CA TCMR treatment remain unclear, as it is a newly described phenomenon. Some researchers treated this ailment in a very similar way to acute TCMR with steroids combined with ATG, so the profile of cardiovascular side-effects should be similar to patients treated for this condition. Yet, different approaches included many more therapeutic agents such as methylprednisolone (4 mg/d), mycophenolate mofetil (500–1500 mg/d), Tacrolimus (minimal concentration 5–8 th/mL), everolimus (minimal concentration 3–8 ng/mL), basiliximab (20 mg in days 0 and 4) and ATG [50,51]. Cardiovascular side-effects upon steroids, MMF and tac have been mentioned already above. Everolimus is an mTOR inhibitor that [113] induces a broad spectrum of adverse effects that might restrict the drug’s clinical applicability. The meta-analysis conducted by Arena et al. revealed that stomatitis, leukopenia, anorexia, anemia and fatigue were the most frequently reported side-effects [114]. Adverse events increasing the cardiovascular risk profile are hypercholesterolemia and hyperglycemia; however, the safety profile of everolimus requires further investigation [114,115].

### 7.4. Borderline Rejection

The BCR therapeutic approach is still a subject of ongoing discussion, which regard mostly its modality, regimen and efficacy [54]. Besides the administration of methylprednisolone and prednisone in a rapid steroid regimen, the induction protocol is based on basiliximab, alemtuzumab or ATG [65].

Basiliximab has been found to be a valuable part of immunosuppressive regimens utilized in diverse solid organ transplantations, owing to its ability to decrease the incidence of acute rejection. Several research studies conducted on post-kidney-transplant patients have unveiled a substantial reduction in acute and chronic cellular rejection incidence, without raising the risk of occurrence of adverse effects [116]. The rationale behind the use of Basiliximab relies on the crucial role in triggering rejection being played by IL-2 [117]. Consequently, it may lead to the potential suspension of steroid administration and reduced dosages of calcineurin inhibitors, thereby diminishing the deleterious side-effects associated with the use of steroids and CNIs, including infections, metabolic abnormalities, cardiovascular complications and nephrotoxicity following transplantation [57,58,118,119].

Among the most common adverse effects in renal transplant recipients, acute and severe hypersensitivity reactions were reported. They manifested themselves by hypotension, tachycardia, cardiac failure, dyspnea, wheezing, bronchospasm, pulmonary oedema, respiratory failure, urticaria, rash, pruritus and/or sneezing, as well as capillary leak syndrome and cytokine release syndrome [60]. In two five-year studies based on a pooled analysis, the incidence and cause of death remained similar in both treatment groups (Basiliximab 15%, placebo 11%). The primary cause of death was cardiovascular disorders (Baxiliximab 5%, placebo 4%), mainly cardiac failure and myocardial infarction [120]. In two double-blind, controlled, multi-center trials, 363 patients underwent therapy with Basiliximab, while 359 were placebo-treated patients; the aim of these trials was to reveal probable patterns of adverse effects. The results have not demonstrated any relevant differences between the two groups. Regarding the cardiovascular effects of the therapy, there were no significant repercussions on the outcome:Tachycardia—Basiliximab 28 (8%) vs. placebo 21 (6%);Hypertension—Basiliximab 97 (27%) vs. placebo 93 (26%);Hypotension—Basiliximab 30 (8%) vs. placebo 38 (11%) [120].

Alemtuzumab significantly facilitates the management of patients in the early post-transplant period, as its use in induction therapy results in reducing the incidence of early acute rejection [61]. Given that it causes long-lasting lymphopenia, which particularly affects both T and B lymphocytes, there is hope that its use may simplify the development of steroid- and calcineurin-free regimens. Furthermore, this therapy is more cost-efficient in comparison to those utilizing other anti-lymphocyte agents [62,63].

Numerous research studies have been conducted in order to determine the influence of Alemtuzumab on the cardiovascular system, as it has been found to induce congestive heart failure, left ventricle (LV) dysfunction, arrhythmias and myocardial ischemia [121,122,123,124]. Although the precise etiology of the adverse effects remains unknown, it seems that alemtuzumab-induced cytokine release syndrome, characterized by an abrupt and excessive release of cytokines, may be a contributing factor. This phenomenon could potentially be mitigated through the utilization of anti-interleukin-6 or tumor necrosis factor inhibitors, which could help temper the cytokine response [123,125].

Despite Alemtuzumab’s promising results as an inductive agent in the studies conducted thus far, further research is needed as a means to prove its efficacy and range of viable indications, as well as to discover effective therapeutic approaches that would prevent the adverse effects from developing.

## 8. Discussion

To prevent grafts from rejection, all kidney transplant recipients are exposed to side-effects related to maintenance immunosuppressive therapy, consisting of CNIs, steroids and MMF.

In patients with ABMR, the risk of cardiovascular complications is particularly high due to endothelial dysfunction and inflammation developing throughout the process and damaging the blood vessels. By consequence, the risk of hypertension, atherosclerosis and other cardiovascular diseases is increased. However, the final manifestation of adverse effects and their implication on the CVS depends on the applied treatment method. The most common guidelines for ABMR treatment recommend the implementation of plasmapheresis and IVIG, both considered to be relatively safe, yet not devoid of negative impact on CVS. Other immunosuppressive agents administered in patients with ABMR may also lead to CVS-related side-effects, such as hypotension, ventricular arrhythmias, hypercoagulability and deep vein thrombosis. Hence, it is crucial to establish an optimal therapeutic scheme for ABMR treatment, being a compromise between the maintenance of immunosuppression and the cardiovascular risk developed due to the therapy.

Patients suffering from TCMR are mainly exposed to steroids-related side effects. By contrast to patients with ABMR, they are possibly at higher risk of developing dyslipidemia states or diabetes mellitus and should be monitored closely for any metabolic disorders. We estimate that cardiovascular risk can be potentially lower in TCMR patients than in ABMR ones, yet it requires further investigation.

Borderline rejection is a challenging condition to diagnose and manage. Besides the steroids, the protocols applied in patients suffering from BCR suggest the administration of basiliximab and alemtuzumab, both of which are considered to be relatively safe when assessing the cardiovascular profile; however, their use and possible side-effects require further research.

Mixed rejection is a complex condition that shares features of both TCMR and ABMR. The treatment of mixed rejection requires a tailored approach combining therapeutic methods used in both TCMR and ABMR. However, their proportions when applied in mixed rejection may vary, depending on the predominant rejection type. By consequence, patients suffering from mixed rejection may be at the higher risk of developing CVS complications. Taking into consideration the analyzed data and numerous research outcomes summarized in this study, managing kidney graft rejection requires a multidisciplinary strategy incorporating not only nephrologists and transplant surgeons but also cardiologists and endocrinologists. It is crucial to promptly detect and address possible cardiovascular risk factors and regularly monitor renal function, blood pressure, lipid profile and glucose metabolism. Moreover, patients who experience graft rejection might need close monitoring and additional therapies, such as dialysis or re-transplantation, to stop further harm to the cardiovascular system and enhance overall treatment outcomes. The findings presented in our review encompass a range of well-examined medications and regimens, as well as emerging concepts in the field of transplantation. It is crucial to recognize that the results of our study should be utilized with care as a primary reference for clinical decision making in patient management. Above all, these findings can serve as a stimulus for researchers to conduct more extensive clinical investigations concerning cardiovascular side-effects during the treatment of kidney rejection. It is important to note that this area of study remains insufficiently explored and necessitates further investigation.

Limitations: While our review article on the therapy in the course of kidney graft rejection and its impact on the cardiovascular system is comprehensive and informative, it is important to acknowledge some limitations inherent in our study. These limitations may influence the generalizability and interpretation of our findings. We would like to outline a few key limitations:-Selection bias: One potential limitation of our study is the possibility of selection bias in the included references. Despite our best efforts to conduct an extensive literature search, it is possible that some relevant studies were inadvertently excluded.-Publication bias: Another potential limitation is the presence of publication bias, whereby studies with positive or significant results are more likely to be published than those with negative or nonsignificant findings. This bias can impact the overall assessment of the therapy’s efficacy and safety.-Heterogeneity of study designs: The studies included in our review may exhibit heterogeneity in terms of study design, patient populations, therapeutic interventions and outcome measures. This heterogeneity could affect the comparability and synthesis of the results, potentially limiting the strength of our conclusions.-Time constraints and knowledge cutoff: Our review article was completed within a specific time frame, and the data collection was limited to the available literature up until May 2023. Newer studies published after this cutoff date may exist, potentially impacting the comprehensiveness of our review.-Potential confounding factors: The impact of therapy on the cardiovascular system in the context of kidney graft rejection can be influenced by various confounding factors, such as comorbidities, concurrent medications and patient characteristics. It is challenging to control for all these factors in observational studies, which could introduce confounding biases that limit the strength of our conclusions.

Despite these limitations, we believe that our review provides a valuable synthesis of the literature, highlighting the importance of therapy in managing kidney graft rejection and its impact on the cardiovascular system.

## Figures and Tables

**Figure 1 life-13-01458-f001:**
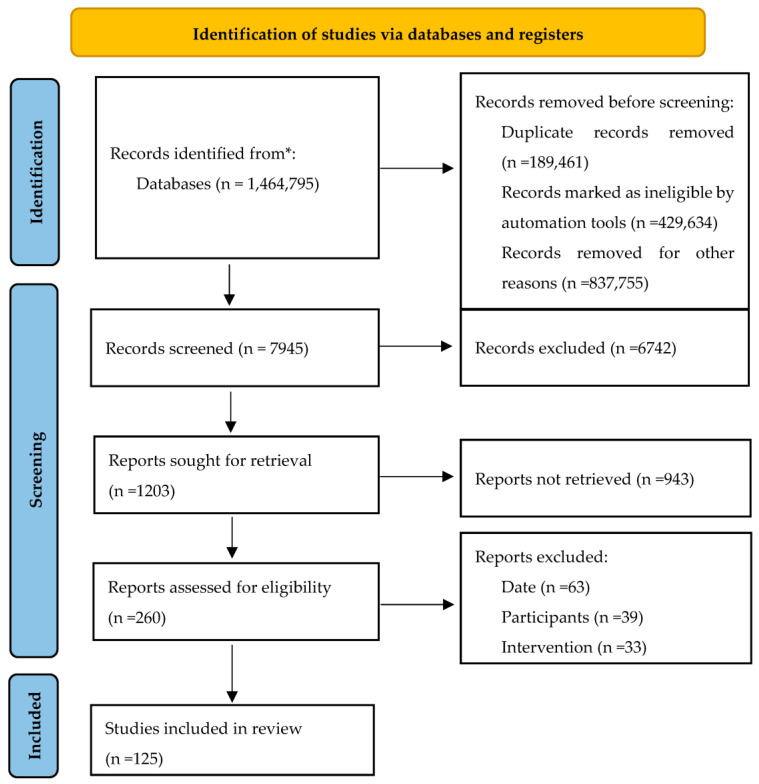
PRISMA flow diagram of the study selection process. * The literature search included Google Scholar and PubMed databases.

**Table 1 life-13-01458-t001:** Active ABMR diagnostic criteria. Abbreviations: ABMR—antibody-mediated rejection, g—glomerulitis, ptc—peritubular capillaritis, v—endarteritis, TCMR—T-cell-mediated rejection, IF—immunofluorescence, IHC—immunohistochemistry, DSA—donor-specific antibody, HLA—human leukocyte antigen.

In order to diagnose Active ABMR, all 3 of the following criteria must be met:	
1.	Histologic evidence of acute tissue injury, including 1 or more of the following: Microvascular inflammation (g > 0 and/or ptc > 0), in the absence of recurrent or de novo glomerulonephritis, although in the presence of acute TCMR, borderline infiltrate or infection, ptc ≥ 1 alone is not sufficient and g must be ≥1; Intimal or transmural arteritis (v > 0) Acute thrombotic microangiopathy, in the absence of any other cause; Acute tubular injury, in the absence of any other apparent cause.
2.	Evidence of current/recent antibody interaction with vascular endothelium, including 1 or more of the following: Linear C4d staining in peritubular capillaries (C4d2 or C4d3 by IF on frozen sections, or C4d > 0 by IHC on paraffin sections); At least moderate microvascular inflammation ([g + ptc] ≥ 2) in the absence of recurrent or de novo glomerulonephritis, although in the presence of acute TCMR, borderline infiltrate or infection, ptc ≥ 2 alone is not sufficient and g must be ≥1; Increased expression of gene transcripts/classifiers in the biopsy tissue strongly associated with ABMR, if thoroughly validated.
3.	Serologic evidence of donor-specific antibodies (DSA to HLA or other antigens). C4d staining or expression of validated transcripts/classifiers as noted above in criterion 2 may substitute for DSA; however, thorough DSA testing, including testing for non-HLA antibodies if HLA antibody testing is negative, is strongly advised whenever criteria 1 and 2 are met

**Table 2 life-13-01458-t002:** Chronic active ABMR diagnostic criteria. Abbreviations: ABMR—antibody-mediated rejection, cg—chronic glomerulopathy, TMA—throbotic microangiopathy, EM—electron microscopy, TCMR—t-cell-mediated rejection, DSA—donor-specific antibody.

In order to diagnose Chronic Active ABMR, all 3 of the following criteria must be met:	
1.	Morphologic evidence of chronic tissue injury, including 1 or more of the following: Transplant glomerulopathy (cg > 0) if no evidence of chronic TMA or chronic recurrent/de novo glomerulonephritis; includes changes evident by electron microscopy (EM) alone (cg1a); Severe peritubular capillary basement membrane multilayering (requires EM); Arterial intimal fibrosis of new onset, excluding other causes; leukocytes within the sclerotic intima favor chronic ABMR if there is no prior history of TCMR, but are not required.
2.	Identical to criterion 2 for active ABMR above.
3.	Identical to criterion 3 for active ABMR above, including strong recommendation for DSA testing whenever criteria 1 and 2 are met. Biopsies meeting criterion 1 but not criterion 2 with current or prior evidence of DSA (post transplant) may be stated as showing chronic ABMR; however, remote DSA should not be considered for diagnosis of chronic active or active ABMR.

**Table 3 life-13-01458-t003:** Chronic (inactive) diagnostic criteria. Abbreviations: ABMR—antibody-mediated rejection, cg—chronic glomerulopathy, DSA—donor-specific antibody.

Chronic (inactive) ABMR	
1.	cg > 0 and/or severe ptcml (ptcml1).
2.	Absence of criterion 2 of current/recent antibody interaction with the endothelium.
3.	Prior documented diagnosis of active or chronic active ABMR and/or documented prior evidence of DSA.

**Table 4 life-13-01458-t004:** Acute TCMR diagnostic criteria. Abbreviations: i—inflammation in non-scarred cortex, t—tubulitis, v—endarteritis.

	i	t	v
Grade IA	Interstitial inflammation involves more than 25% of non-sclerotic cortical parenchyma	Moderate tubulitis involving at least 1 tubule excluding severely atrophic tubules	-
Grade IB	Interstitial inflammation involves more than 25% of non-sclerotic cortical parenchyma	Severe tubulitis involving at least 1 tubule excluding severely atrophic tubules	-
Grade IIA	Interstitial inflammation can be present but does not have to	Tubulitis can be present but does not have to	Mild to moderate intimal arteritis
Grade II B	Interstitial inflammation can be present but does not have to	Tubulitis can be present but does not have to	Severe intimal arteritis
Grade III	Interstitial inflammation can be present but does not have to	Tubulitis can be present but does not have to	Transmural arteritis or/and arterial fibrinoid necrosis, which involve medial smooth muscle together with mononuclear cell intimal arteriris

**Table 5 life-13-01458-t005:** Chronic active TCMR diagnostic criteria. Abbreviations: i-IFTA—inflammation in scarred cortex, ti—total cortical inflammation, t—tubulitis, cv—arterial intimal fibrosis.

	i-IFTA	ti	t	cv
Grade IA	Interstitial inflammation involves not less than 25% of sclerotic cortical parenchyma	Interstitial inflammation involves not less than 25% of total cortical parenchyma	Moderate tubulitis involving at least 1 tubule excluding severely atrophic tubules	-
Grade IB	Interstitial inflammation involves not less than 25% of sclerotic cortical parenchyma	Interstitial inflammation involves not less than 25% of total cortical parenchyma	Severe tubulitis involving at least 1 tubule excluding severely atrophic tubules	-
Grade II				Chronic allograft arteriopathy including arterial intimal fibrosis together with mononuclear cell inflammation in fibrosis. Moreover, the formation of neointima

**Table 6 life-13-01458-t006:** Borderline cellular rejection diagnostic criteria. Abbreviations: BCR—borderline cellular rejection, t—tubulitis, i—inflammation in non-scarred cortex, v—endarteritis.

To diagnose BCR, all 3 of the following criteria must occur:	
1.	Foci of tubulitis (t1, t2 or t3).
2.	Mild interstitial inflammation (i1), or mild (t1) tubulitis with moderate-severe interstitial inflammation (i2 or i3).
3.	No intimal or transmural arteritis (v = 0).

## Data Availability

Not applicable.

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
