# Peer review of "Therapy in the Course of Kidney Graft Rejection—Implications for the Cardiovascular System—A Systematic Review"

_life, 2023, doi:10.3390/life13071458_

Round 1
Reviewer 1 Report
1. The title is misleading - the graft rejection may lead to the graft failure, but it is not a synonym - please, change "failure" into "rejection"
2. Line 713: is anemia really a vascular disorder?
3. The Authors attempted to summarize the cardiovasular side effects of renal graft rejection therapy, but in the text they discuss also complications from other systems (e.g. lines 853-855). I suggest to focus on a scheduled issues, and not to widen the topic (or to change the title accordingly).
4. I would expect that the Authors present limitations of their study in the Discussion.
Author Response
Dear Reviewer,
Thank you for taking the time to review our article. We appreciate your careful assessment of our work and your valuable feedback.
As suggested, we provided the following changes to our article:
- Replaced the word "failure" into "rejection" in the title of the article.
- Revised the article for inaccuracies regarding the categories of adverse effects and removed wide descriptions of others than cardiovascular ones.
- Same comment as in point 2.
- Limitations of the study provided after the discussion.
Once again, we sincerely appreciate your thoughtful evaluation of our manuscript. In case you have any additional comments or suggestions, please feel free to contact us.
Best regards,
Jakub Mizera
Reviewer 2 Report
Dear Authors:
Thank you for coming up with this paper and providing extensive information and analysis regarding post kidney transplant rejections. which is a very complicated topic. I read your paper with a great interest. I found it very informative, and it would be a great resource for the clinicians and medical educators in the field of transplantation. Specially with classification of the kidney rejection and how to manage them differently.
I did not notice any major flows and mistakes regarding English grammar.
Author Response
Dear Reviewer,
Thank you for taking the time to review our article. We appreciate your careful assessment of our work and your valuable feedback. As suggested, minor editing of English language was provided. Should you have any additional comments or suggestions, please feel free to contact us.
Best regards,
Jakub Mizera